# The Probiotic VSL#3^®^ Does Not Seem to Be Efficacious for the Treatment of Gastrointestinal Symptomatology of Patients with Fibromyalgia: A Randomized, Double-Blind, Placebo-Controlled Clinical Trial

**DOI:** 10.3390/ph14101063

**Published:** 2021-10-19

**Authors:** Elena P. Calandre, Javier Hidalgo-Tallon, Rocio Molina-Barea, Fernando Rico-Villademoros, Cristina Molina-Hidalgo, Juan M. Garcia-Leiva, Maria Dolores Carrillo-Izquierdo, Mahmoud Slim

**Affiliations:** 1Instituto de Neurociencias, Universidad de Granada, 18100 Granada, Spain; fjht63@gmail.com (J.H.-T.); fernando.ricovillademoros@gmail.com (F.R.-V.); jmgleiva@ugr.es (J.M.G.-L.); 2Servicio de Cirugía, Complejo Hospitalario de Jaen, 23007 Jaén, Spain; barea1984@gmail.com; 3Departamento de Psicología Médica, Universidad de Granada, 18011 Granada, Spain; cristinamolinapsico@gmail.com; 4Departamento de Enfermería, Universidad Católica de Murcia, 30107 Murcia, Spain; mariadocarrillo@gmail.com; 5Division of Neurology, The Hospital for Sick Children, Toronto, ON M5G 1X8, Canada; mahmoud.slim@gmail.com

**Keywords:** fibromyalgia, gastrointestinal symptoms, probiotic, VSL#3^®^, efficacy, tolerability

## Abstract

Gastrointestinal symptomatology is frequent among patients with fibromyalgia, which increases disease burden and lacks specific treatment, either pharmacological or non-pharmacological. We aimed to evaluate the efficacy and tolerability of a multi-strain probiotic, VSL#3^®^, for the treatment of fibromyalgia-associated gastrointestinal manifestations. This randomized, placebo-controlled trial included 12 weeks of probiotic or placebo treatment followed by 12 weeks of follow up. The primary outcome variable was the mean change from the baseline to the endpoint in the composite severity score of the three main gastrointestinal symptoms reported by patients with fibromyalgia (abdominal pain, abdominal bloating and meteorism). Secondary outcome variables were the severity of additional gastrointestinal symptoms, fibromyalgia severity, depression, sleep disturbance, health-related quality of life and patients’ overall impression of improvement. No differences were found between VSL#3^®^ (n = 54) and the placebo (n = 56) in the primary outcome (estimated treatment difference: 1.1; 95% confidence interval [CI]: −2.1, 4.2; *p* = 0.501), or in any of the secondary outcomes. However, responders to VSL#3 were more likely to maintain any improvement during the follow-up period compared to responders in the placebo arm. Overall, VSL#3 tolerability was good. Our data could not demonstrate any beneficial effects of VSL#3^®^ either on the composite score of severity of abdominal pain, bloating and meteorism or in any of the secondary outcome variables. More research is needed to elucidate specific factors that may predict a favourable response to treatment in patients with fibromyalgia.

## 1. Introduction

Fibromyalgia is a complex syndrome in that, although its main characteristic is chronic generalized musculoskeletal pain, this is accompanied in most patients by other symptoms, the most common of which are non-restorative sleep, chronic fatigue, cognitive difficulties and anxious and/or depressive symptoms [1]. It is included within the central sensitization syndromes, such as migraine, irritable bowel syndrome (IBS) or temporomandibular disorders, with which it shows a high comorbidity [2].

Gastrointestinal symptoms are very common in patients with fibromyalgia, and could be derived from the presence of comorbid IBS or other underlying pathophysiological mechanisms [3]. A systematic review reported a pooled prevalence of 51% for functional gastrointestinal disorders and 46% for IBS among patients with fibromyalgia [4]. On the other hand, even among patients with fibromyalgia who do not meet the criteria to diagnose IBS, the presence of gastrointestinal symptomatology is frequently observed [3]. The cause of these symptoms remains unknown, although some studies suggest that they could be due to intestinal bacterial overgrowth or intestinal permeability alterations [5,6,7]. A recent study in patients with chronic fatigue syndrome, a pathology that shows a broad overlap with fibromyalgia, described an increased likelihood of intestinal dysbiosis for these patients [8]. Patients with fibromyalgia also showed an alteration in gut microbiota, although the role of these alterations should be further elucidated [9,10].

No specific treatment for alleviating the gastrointestinal symptoms associated with fibromyalgia has been studied, despite their frequency and most patients describing them as extremely annoying [3,11]. This may explain the frequency with which these patients resort to different types of diets, even though their benefits have not been previously demonstrated [12,13,14]. In this regard, the use of probiotics, alone or associated with prebiotics (synbiotics), could be an interesting therapeutic approach for managing gastrointestinal symptoms in patients with fibromyalgia. Probiotics have been studied in a variety of clinical conditions, including gastrointestinal disorders, dermatological disorders and metabolic diseases [15,16]. Among gastrointestinal disorders, the most studied condition is IBS, where probiotics seem to exert a favourable response in global symptoms, although there are not still enough data to specify which individual probiotics could be more effective [17]. Probiotics seem to be well tolerated in general [17].

Considering the frequent presence of gastrointestinal symptoms in patients with fibromyalgia who might be susceptible to treatment with probiotics, the objective of this trial was to assess the efficacy and tolerability of VSL#3^®^, a multi-strain probiotic, which has demonstrated a trend for an overall improvement in the treatment of IBS [18,19] in patients with fibromyalgia and gastrointestinal symptomatology.

## 2. Results

### 2.1. Patient Disposition and Characteristics

One hundred and ten patients were recruited from May 2018 to November 2019 and allocated to either placebo (n = 56) or VSL#3^®^ (n = 54). Twenty-five subjects (44.6%) in the placebo group and 28 (51.8%) subjects in the VSL#3^®^ group did not complete the study (Figure 1). Fifty-three subjects in each study group were included in the analysis of the primary outcome and that of a proportion of responders according to the composite score of abdominal pain, bloating and meteorism; secondary efficacy outcomes, including the proportion of responders according to the Patient Global Improvement scale (PGI), were evaluated in 35 subjects in the placebo group and 28 subjects in the VSL#3^®^ group. All randomised subjects were included in the safety analysis.

Subjects were middle-aged, and the vast majority were women (Table 1). Comorbidity was high, with anxiety/depressive disorder, tension-type headache, craniomandibular dysfunction, chronic fatigue syndrome and irritable bowel syndrome present in over 50% of the patients (Table 1). Most patients were receiving pharmacological treatment with some activity for the symptoms of fibromyalgia. Benzodiazepines, antidepressants, NSAIDs, paracetamol and tramadol were mainly used, each in over 30% of the patients; one-third of patients were receiving gastroprotectant drugs (Table 1). At baseline, the study groups were generally well-balanced regarding demographics and clinical characteristics (Table 1), including the individual gastrointestinal symptoms of abdominal pain, abdominal bloating, meteorism, and the composite score of these three symptoms (Table 2). However, the impact of fibromyalgia as evaluated with the revised fibromyalgia impact questionnaire (FIQR) was greater among placebo-treated patients than in VSL#3^®^-treated patients (FIQR total score 75.5 ± 12.3 vs. 70.0 ± 17.8), although the difference was not statistically significant.

### 2.2. Primary Outcome

In the intent-to-treat last observation carried forward (LOCF) analysis, at week 12, the severity of pain, bloating and meteorism as measured with the composite score was reduced by 6.5 points among VSL#3^®^-treated patients and 5.4 points among placebo-treated patients; this difference was not statistically different (estimated treatment difference (ETD): 1.1; 95% confidence interval [CI]: −2.1 to 4.2; *p* = 0.501) nor clinically relevant (Cohen’s d = 0.13).

### 2.3. Secondary Outcomes

#### 2.3.1. Gastrointestinal Symptoms

There were no statistically significant differences between VSL#3^®^ and placebo in any of the individual gastrointestinal symptoms (Table 2). The largest difference was observed in diarrhoea, in favour of VSL#3^®^ (ETD: 1.3; 95% CI: −0.4 to 2.9; *p* = 0.131). All effect sizes for the differences between VSL#3^®^ and placebo in the gastrointestinal symptoms were trivial, except for a small effect size for abdominal pain and diarrhoea in favour of VSL#3 and a small effect size for constipation in favour of the placebo. Results of the complete case analysis for the primary outcome and the gastrointestinal symptoms were generally similar to those of the LOCF approach, except for the largest mean within-group changes in the scores (Table 3).

In the intent-to-treat population and using an LOCF approach, by week 12, 27 out of the 53 (50.9%) patients showed a reduction equal to or greater than 30% in the composite score of abdominal pain, bloating and meteorism in the VSL#3^®^ group, compared to 22 of the 53 (41.5%) in the placebo group (relative risk [RR]: 1.23; 95% CI: 0.81 to 1.86). The proportion of responders according to the PGI was 22.2% and 26.4% for VSL#3- and placebo-treated patients, respectively (RR: 0.86; 95% CI: 0.44 to 1.68) (Figure 2).

Among the patients who responded to treatment at week 12 according to the reduction in the composite score of abdominal pain, bloating and meteorism, after discontinuing the study treatment, the composite score increased by over four points during the 12-week follow-up extension in the placebo group and by over one point in the VSL#3^®^ group, with an ETD of 2.8 points (95% CI: 0.0 to 5.6; *p* = 0.048) (Figure 3).

There were no relevant differences in the baseline characteristics between responders and non-responders in either the total sample or in the VSL#3^®^ and placebo groups (data not shown).

#### 2.3.2. The Effect on Other Symptoms of Fibromyalgia and Quality of Life

Overall, the severity of fibromyalgia was reduced in both study groups, but to a greater extent among placebo-treated patients, although the differences between the two study groups were not statistically significant (ETD: −5.2; 95% CI: −12.0 to 1.6; *p* = 0.128; Cohen’s d: 0.40). Except for stiffness, which improved to a significantly greater extent with the placebo than with VSL#3^®^ (ETD: −1.5; 95% CI: −2.8 to 0.1; *p* = 0.0304; Cohen’s d: 0.56), there were no significant differences in the changes from baseline in the core symptoms of fibromyalgia, sleep impairment, depressive symptoms or quality of life between VSL#3^®^ and the placebo (Table 4).

### 2.4. Tolerability

One-third of the patients in each study group reported at least one adverse event. Seven (13.0%) patients in the VSL#3^®^ group and six (10.7%) in the placebo group discontinued the treatment due to adverse events. The vast majority of the adverse events were gastrointestinal related, with some differences between the two study groups in the adverse event profile. Abdominal distension was more frequent among VSL#3^®^-treated patients, whereas upper abdominal pain was more frequent among placebo-treated patients; however, none of the differences was statistically significant (Table 5).

## 3. Discussion

Overall, our data could not demonstrate any beneficial effects of VSL#3^®^ either on the composite score of severity of abdominal pain, bloating and meteorism or in any of the secondary outcome variables. This lack of benefit can be potentially attributed to several factors, including the elevated placebo response, the high proportion of patients who withdrew from the study, and the presence of rather complicated mechanisms underlying the gastrointestinal manifestations in fibromyalgia.

The relevance of the placebo effect in fibromyalgia clinical trials is substantial and has been investigated in several systematic reviews and meta-analyses [20,21,22]. It has been estimated that the mean placebo effect for pain reduction in patients with fibromyalgia is 30.8% when considering a 30% pain reduction, and 18.8% when considering a 50% pain reduction [20,21]. A recent meta-analysis found that, in relation to patients that received no treatment, fibromyalgia patients receiving placebo experienced significant improvement not only in pain, but also in fatigue, sleep quality, physical function and FIQ total score [22]. In our study, the proportion of placebo responders for the main outcome variable was 50.9%. Similar placebo effect rates have been described in clinical trials evaluating probiotics in patients with IBS. In their review, Rogers and Mousa indicated the presence of a high placebo effect among patients with IBS ranging between 30% and 50%. Several mediators of the placebo effect, particularly in patients with functional somatic disorders, have been suggested, including Pavlovian conditioning, belief outcomes, and patient expectations, among other factors [23].

The dropout rate was also disproportionately high. Nocebo effect is also very relevant in fibromyalgia clinical trials, and it has been estimated to represent between 9% and 11% of patient dropouts [21,24]. Consistent with these estimations, the percentage of placebo-treated patients in our study that withdrew due to tolerability issues was 10.7% of the sample, slightly less than the dropout rate in the VSL#3^®^ group, which was of 12%. However, 17 (31.5%) patients in the VSL#3^®^-treated group and 15 (26.8%) in the placebo-treated group withdrew due to reasons unrelated to tolerability and/or efficacy issues, mainly loss of follow up; the percentage of withdrawals was similar across participating centres.

The effects of probiotics on human health seem to be related to different effects, such as a decrease in inflammation, decrease in intestinal permeability, modification of the intestinal microbiota, and metabolism modulation. These effects are mediated by multiple mechanisms of action, including the colonization and normalization of perturbed intestinal microbial communities, competitive exclusion of pathogens, modulation of enzymatic activities and production of volatile fatty acids. Nevertheless, it is important to highlight that the mechanisms underlying the exacerbation of gastrointestinal manifestations in fibromyalgia appear to be far more complex and extend beyond the possible small intestinal bacterial overgrowth, gut microbiota alterations or symbiosis [3]. Therefore, this may provide another possible explanation for the lack of benefit of VSL#3 on the primary efficacy outcome. More research is needed to further understand the specific patient characteristics that may predict a favourable response to VSL#3^®^.

Interestingly, VSL#3-treated patients who were considered as responders to treatment according to the primary outcome variable maintained the degree of improvement obtained after the treatment period during the follow-up period, whereas in placebo-treated patients who were considered as responders, the improvement decreased during the follow-up period. This suggests that at least a subgroup of patients obtained a benefit from VSL#3^®^ treatment. Unfortunately, we were not able to identify any characteristic that could differentiate placebo- from VSL#3^®^-responders.

In the last five years, the efficacy of probiotics in the treatment of IBS has been evaluated in several systematic reviews and meta-analyses [17,25,26,27]. These reviews reached a common conclusion that probiotics seem to be beneficial for IBS symptoms and that their tolerability is generally good, although more information is needed in relation to probiotic type, probiotic dosage and treatment length. With one exception [27], they also agree in considering that multi-strain probiotics seem to be preferable over single-strain probiotics.

The use of probiotics in the management of IBS has been recently revised by the American Gastroenterological Association in a technical review that found that, although data concerning the potential efficacy of probiotics on the management of IBS are substantial, no single strain or combination has been studied in a sufficiently rigorous manner [28]. For this reason, the American Gastroenterological Association advocates the use of probiotics for the treatment of IBS only in the context of a clinical trial [29].

VSL#3^®^ has been the object of two meta-analyses in the treatment of IBS. The first one, published in 2018, evaluated the efficacy and tolerability of VSL3#3^®^ for the treatment of IBS [18]; the authors concluded that, although a trend for global overall improvement was observed, no significant differences with placebo were found for specific symptoms such as abdominal pain, bloating or stool consistency. Probiotic-associated side effects were detailed only in one of the five clinical trials included in the meta-analysis and reported a more frequent worsening of the gastrointestinal symptoms in VSL#3^®^-treated patients than in placebo-treated patients. In our study, almost all side-effects reported by patients who received VSL#3^®^ were also related to the worsening of the previous gastrointestinal symptoms (Table 5). We would like to note that some authors have reported that the formulation of VSL#3 used in the studies conducted prior to 2016 is not the same as the one used here; thus, the results reported in this meta-analysis could be referred to that formulation and not the one we used in our study [30]. The second meta-analysis was based on the tolerability of VSL#3^®^ in any clinical condition, which included IBS, obesity, ulcerative colitis, and early menopause, concluding that the safety profile of VSL#3^®^ was not significantly different from the placebo, and was similar to that of other probiotics [19]. However, there are uncertainties about this meta-analysis because the actual number of patients examined was too small and the pathologies and the probiotic dosages too heterogeneous.

To the best of our knowledge, only one study has been published evaluating the use of probiotics in the treatment of fibromyalgia [31]. The objective of this randomised, placebo-controlled trial, which also used a multi-strain probiotic, was to investigate the potential efficacy of the probiotic on the cognition, emotional symptoms and functional state of the patients. Thus, we cannot establish any comparison in relation to our primary objective, which was to assess the gastrointestinal symptomatology of the patients. However, the authors assessed other variables that we also evaluated, such as depression, anxiety, fibromyalgia pain and impact and health-related quality of life, As in our case, no significant differences were found between the probiotic and placebo in relation to any of these outcomes.

Our study has some limitations. The high dropout rate and its impact on the study estimates because of the missing data as well as the placebo effect in our study were relatively high, prompting cautious interpretation of the study findings. Although we performed a secondary analysis using a complete case approach in order to limit the influence of the imputation method to handle missing data, it is important to bear in mind that complete case analysis is appropriate only when the participants in the analysis can be regarded as a random sample of the study population (i.e., when the missing mechanism is missing completely at random) [32], which cannot be assumed to be the case in our study; in addition, complete case analysis tends to overestimate treatment effects. Therefore, complete case analysis can only be considered as a sensitivity analysis. In addition, due to the lack of a validated scale for measuring our primary outcome, we had to use an ad hoc instrument to assess the severity of gastrointestinal symptoms. Additionally, the lack of sample size calculation due to the absence of published data on the primary outcome measure may have prevented us from adequately controlling the power in the current study. Finally, we did not investigate the composition of patients’ microbiota either at the beginning or the end of the trial; this would have been a worthwhile approach, since different experimental and clinical studies have shown that multi-train probiotics can improve health by modifying the gut microbiota composition [33,34,35,36].

## 4. Materials and Methods

### 4.1. Study Design

In this study, a randomised double-blind placebo-controlled trial evaluating the efficacy and tolerability of VSL#3^®^ in the treatment of patients with fibromyalgia and associated gastrointestinal symptomatology was conducted. VSL#3^®^ (manufactured for Actial Farmaceutica Srl) is a high-concentration multi-strain probiotic mix, commercially available in 450 billion CFU/sachet, containing the following: (i) one strain of *Streptococcus thermophilus BT01*; (ii) three strains of Bifidobacteria: *B. breve BB02*, *B. animalis subsp. lactis BL03* (previously identified as *B. longum BL03*) and *B. animalis subsp. lactis BI04* (previously identified as *B. infantis BI04*); and (iii) four strains of Lactobacilli: *L. acidophilus BA05*, *L. plantarum BP06*, *L. paracasei BP07* and *L. helveticus BD08* (previously identified as *L. delbrueckii subsp. bulgaricus*
*BD08*) [37]. The composition of the placebo was maltose, cornstarch and silicon dioxide

The treatment was administered during a 12-week period, and the participants were followed for an additional 12-week period in order to follow evolution after treatment. The trial protocol was approved both by the Biomedical Research Ethics Committee of the province of Granada (Granada, Spain) and by the Ethics Committee of the Catholic University of Murcia (Murcia, Spain), the two cities where the trial was carried out. Written informed consent was obtained from every subject before inclusion in the study. The trial was registered at ClinicalTrials.gov with the identifier NCT04256785.

### 4.2. Participants

Patients were recruited from several fibromyalgia associations who regularly attended the two outpatient clinics where the trial was performed.

The inclusion criteria were the following: (a) diagnosis with fibromyalgia, confirmed at the screening of patients using the ACR 2016 criteria [38]; (b) 18 years of age or older; (c) agreement to voluntarily participate in the study by signing informed consent; (d) willingness to, with no need under medical criteria, maintain the treatment previously received for fibromyalgia, both of pharmacological and non-pharmacological types, with no change in life habits especially regarding habitual diet during the trial’s duration; and (e) regular suffering (two or more times per week) from three or more of the following symptoms: abdominal pain, abdominal bloating, meteorism, flatulence, nausea, dyspepsia, eructation, constipation and/or diarrhoea.

The exclusion criteria were as follows: (a) suffering from severe mental illness other than major depression; (b) suffering from severe renal, hepatic or cardiovascular organic disease that, at the discretion of the investigator, could have interfered with participation in the study; (c) suffering from any chronic gastrointestinal disease other than IBS, such as inflammatory bowel disease, active gastroduodenal ulcer or colorectal carcinoma; and (d) pregnancy or breastfeeding. All of the mentioned diseases were required to have been diagnosed by a physician.

### 4.3. Study Assessments

The severity of the following types of gastrointestinal symptoms was evaluated using a 10-point Visual Analogue Scale (VAS): abdominal pain, abdominal bloating, meteorism, flatulence, constipation, diarrhoea, nausea, eructation and dyspepsia.

Secondary assessments were the following:(a)The Revised Fibromyalgia Impact Questionnaire (FIQR) [39]: This instrument was created to assess the overall symptoms related to fibromyalgia. The total score of the FIQR ranges from 0 to 100, and the higher the score, the greater the severity of fibromyalgia. The validated Spanish version was used [40].(b)The 9-item Patient Health Questionnaire (PHQ-9): The objective of this questionnaire is to evaluate depressive symptoms. Its total score ranges from 0 to 27 points; the higher the score, the greater the severity of the depression. Since depression is also a symptom frequently associated with fibromyalgia, it was used to check whether an eventual improvement in gastrointestinal symptoms is reflected in an improvement in depressive symptomatology. A validated Spanish version of the questionnaire was used [41].(c)The Insomnia Severity Inventory (ISI): This is a brief questionnaire which assesses the severity of insomnia. Its total score ranges from 0 to 28 points; the higher the score, the greater the severity of insomnia. The validated Spanish version of the questionnaire was used [42].(d)The Short-Form Health-Survey SF-36: This multi-item generic health survey aims to evaluate general health concepts not specific to any age, disease or treatment group and measures eight health domains: physical functioning, physical role limitations, bodily pain, general health perceptions, vitality, social functioning, emotional limitations and mental health. These domains yield two summary measures: the Physical Component Summary (PCS) and the Mental Component Summary (MCS). The validated Spanish version was applied [43].(e)A seven-point, Likert-type scale, the Patient Global Improvement Scale, was used to assess the relief of patients’ general symptomatology.

### 4.4. Procedure

At the time of screening, demographic and clinical data from each patient were collected, and the fibromyalgia diagnosis was confirmed. Then, each patient was allocated either to VSL#3^®^ or the matching placebo; the treatment was administered as two sachets of study products twice a day for twelve consecutive weeks. Each sachet of VSL#3^®^ contained 450 billion CFU of live freeze-dried bacteria in powder form (Lot. No. 709002, 709003, 802112, 802113). Patients were randomised in a 1:1 ratio to one of the two treatment groups using a random number generator.

On the day of initiation of treatment, the following questionnaires were administered: VAS of abdominal pain, abdominal bloating, meteorism, flatulence, constipation, diarrhoea, nausea, eructation and dyspepsia; FIQR; ISI; PHQ-9; PGI; and SF-36.

Visual analogue scales of gastrointestinal symptomatology were filled in weekly by the patients during the first 4 weeks of the trial and every 2 weeks between weeks 4 and 12 of the trial. At week 12, FIQR, ISI, PHQ-9 and SF-36 were also completed; PGI was filled in on weeks 4, 8 and 12.

At the end of the treatment period, patients entered into a follow-up period and were monitored at 4, 12 and 24 weeks thereafter; in these visits, the VAS of gastrointestinal symptoms, FIQR, PGI and SF-36 were completed.

Adverse effects potentially associated with treatment were collected at each visit through an open-ended question system. During the 12 weeks of treatment, the medication packages were collected to control therapeutic compliance.

### 4.5. Statistical Analysis

Given the absence of previous intervention studies in this area and, in general, limited information on this aspect of fibromyalgia, this was considered a pilot study. Thus, the calculation of the sample size was based on the feasibility of recruiting them. The recruitment of 110 patients was estimated as a reasonably attainable goal considering the volume of patients attending each one of the two participating centres.

The primary outcome variable was the mean change from baseline to endpoint in the composite score of the three main gastrointestinal symptoms reported by patients with fibromyalgia, i.e., abdominal pain, abdominal bloating and meteorism, as evaluated with the 10-point VAS. We selected the primary outcome variable considering the most frequent gastrointestinal symptoms previously observed in patients with fibromyalgia [3], which are also the most common ones in IBS. Secondary outcomes were the mean changes from baseline to endpoint in the scores of the FIQR, ISI, PHQ-9 and SF-36. In addition, the proportion of responders regarding gastrointestinal symptoms was calculated in two ways: the proportion of patients with a reduction equal to or greater than 30% in the composite score of abdominal pain, bloating and meteorism, and the proportion of patients who were highly or very highly improved (i.e., a score of 1 or 2) according to the PGI.

All patients who had a postbaseline evaluation were included in the efficacy analyses, and missing data were imputed using the LOCF approach. A complete case analysis was also performed for the analysis of the mean changes in the scores of the gastrointestinal symptoms. The results were analysed by applying Student’s t-test to independent samples in order to compare the data between the subjects who received the placebo and those who received the active product, as well as to compare the data in the subgroups of patients treated with the placebo and with VSL#3^®^. The proportion of responders and other categorical variables were compared using the χ^2^ test or Fisher’s exact test, as appropriate. Values lower than 0.05 were considered significant. Effect sizes were calculated using Cohen’s d and interpreted as trivial if they were <0.2, small if they were between 0.2 and <0.5, medium if they were between 0.5 and <0.8 and large if they were ≥0.8. All analyses were performed using SPSS version 22.

## 5. Conclusions

In summary, although VSL#3^®^ displayed favourable safety and tolerability profiles in patients with fibromyalgia, it did not improve their gastrointestinal or fibromyalgia symptomatology compared to the placebo. However, the maintenance of the benefit among VSL#3^®^ responders and not among placebo responders suggests that some patients could benefit from treatment with this probiotic. More research is still needed to further elucidate the specific factors that may predict a favourable response to treatment with VSL#3^®^ in patients with fibromyalgia.

## Figures and Tables

**Figure 1 pharmaceuticals-14-01063-f001:**
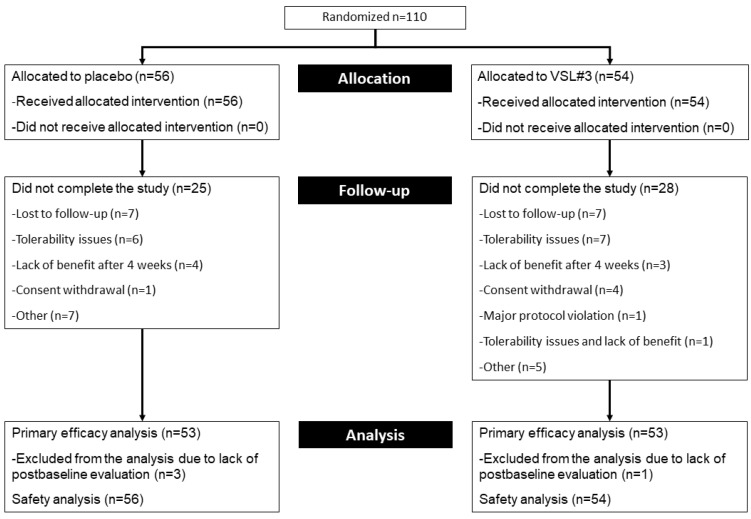
Disposition of trial participants.

**Figure 2 pharmaceuticals-14-01063-f002:**
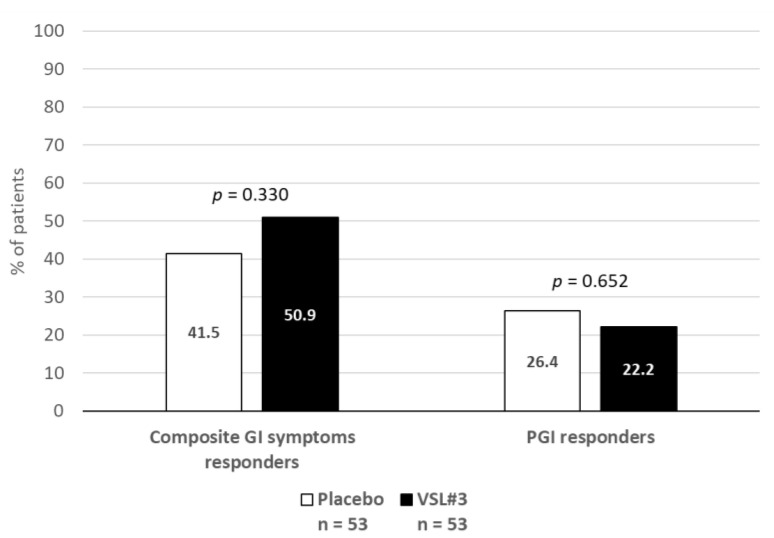
Proportion of responders to treatment.

**Figure 3 pharmaceuticals-14-01063-f003:**
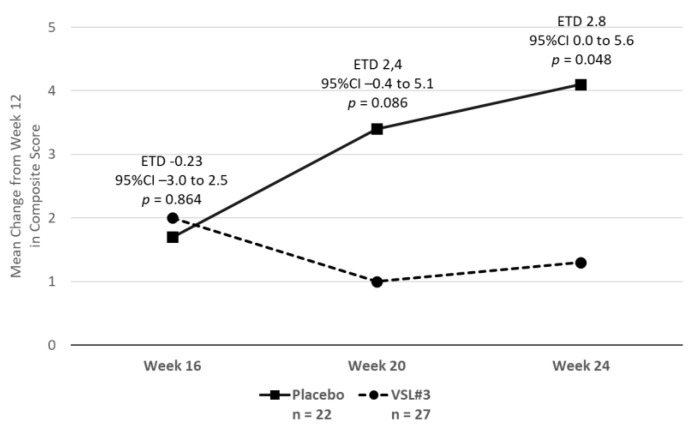
Composite score of abdominal pain, bloating and meteorism after discontinuing the study treatment.

**Table 1 pharmaceuticals-14-01063-t001:** Demographic and clinical characteristics.

Variable	Placebo N = 56	VSL#3^®^ N = 54
Age (years), mean (SD)	55.5 (8.6)	56.0 (7.5)
Sex (females), n (%)	55 (98.2)	52 (96.3)
Weight (kg),	71.2 (13.4)	73.3 (17.7)
Comorbidities ^a^, n (%)		
Anxiety/depressive disorder	48 (85.7)	45 (83.3)
Tension-type headache	40 (71.4)	36 (66.7)
Craniomandibular dysfunction	36 (64.3)	36 (66.7)
Chronic fatigue syndrome	35 (62.5)	28 (51.9)
Irritable bowel syndrome	33 (58.9)	32 (59.3)
Migraine	24 (42.9)	30 (55.6)
Hypothyroidism	25 (44.6)	15 (27.8)
Osteoarthritis	21 (37.5)	15 (27.8)
Rheumatoid arthritis	10 (17.9)	10 (18.5)
Hypercholesterolemia	4 (7.1)	9 (16.7)
Hypertension	9 (16.1)	9 (16.7)
Diabetes mellitus	7 (12.5)	4 (7.4)
Fibromyalgia diagnosis, mean (SD)		
Widespread Pain Index (WPI) [range 0–19]	16.5 (2.6)	15.9 (3.0)
Symptom Severity Score (SSS) [0–12]	9.7 (1.7)	9.3 (2.0)
Fibromyalgia Score (WPI + SSS) [0–31]	26.2 (3.5)	25.3 (4.3)

^a^ Those with a frequency equal to or greater than 10% in any of the trial arms. SD, standard deviation.

**Table 2 pharmaceuticals-14-01063-t002:** Comparison of the impact of VSL#3^®^ and placebo on gastrointestinal symptoms (ITT-LOCF analysis).

	Baseline Mean ± SD		Mean Change (±SD) from Baseline to Week 12		Treatment Difference (Placebo Minus VSL#3)			
Gastrointestinal Symptom	Placebo N = 56	VSL#3^®^ N = 54	Placebo N = 53	VSL#3^®^ N = 53	ETD	95% CI	*p*-Value	Cohen’s d
Primary outcome: composite score of Pain + Bloating + Meteorism	20.9 ± 5.6	20.7 ± 5.0	−5.4 ± 6.6	−6.5 ± 9.5	1.1	−2.1 to 4.2	0.501	0.13
Abdominal pain	6.2 ± 2.5	6.1 ± 2.5	−1.6 ± 3.2	−2.4 ± 3.8	0.8	−0.5 to 2.2	0.228	0.24
Abdominal bloating	7.8 ± 2.2	7.4 ± 2.1	−2.1 ± 2.9	−2.1 ± 3.6	−0.0	−1.3 to 1.3	0.976	0.01
Meteorism	6.9 ± 2.8	7.2 ± 2.5	−1.7 ± 2.8	−2.0 ± 3.5	0.3	−1.0 to 1.5	0.668	0.08
Flatulence	6.5 ± 2.9	6.0 ± 2.3	−1.3 ± 3.2	−1.1 ± 4.0	−0.2	−1.6 to 1.2	0.788	0.05
Constipation	6.7 ± 3.5	6.1 ± 3.6	−2.4 ± 3.8	−1.6 ± 4.2	−0.8	−2.3 to 0.8	0.323	0.20
Diarrhoea	2.6 ± 3.5	4.6 ± 3.9	−1.8 ± 3.8	−3.1 ± 4.7	1.3	−0.4 to 2.9	0.131	0.30
Nausea	3.3 ± 3.2	3.3 ± 3.3	−2.4 ± 3.4	−1.9 ± 4.0	−0.5	−2.0 to 0.9	0.468	0.14
Vomiting	0.7 ± 1.8	0.8 ± 2.3	−0.5 ± 1.5	−0.6 ± 1.9	0.2	−0.5 to 0.8	0.645	0.09
Belching	4.2 ± 3.2	4.1 ± 3.5	−0.6 ± 3.3	−1.1 ± 2.8	0.4	−0.8 to 1.6	0.487	0.14
Dyspepsia	6.2 ± 3.0	6.5 ± 2.9	−2.7 ± 3.8	−3.2 ± 3.5	0.5	−0.9 to 1.9	0.510	0.13

CI, confidence interval; ETD, estimated treatment difference (positive figures favour VSL#3); ITT, intention-to-treat; FMS, fibromyalgia syndrome; LOCF, last observation carried forward; SD, standard deviation.

**Table 3 pharmaceuticals-14-01063-t003:** Comparison of the impact of VSL#3^®^ and placebo on gastrointestinal symptoms (complete case analysis).

	Baseline Mean ± SD		Mean Change (±SD) from Baseline to Week 12		Treatment Difference (Placebo Minus VSL#3)			
Gastrointestinal Symptom	Placebo N = 56	VSL#3^®^ N = 54	Placebo N = 53	VSL#3^®^ N = 53	ETD	95%CI	*p*-Value	Cohen’s d
Primary outcome: composite score of Pain + Bloating + Meteorism	20.9 ± 5.6	20.7 ± 5.0	−7.6 ± 6.1	−7.5 ± 8.4	−0.09	−3.7 to 3.5	0.959	0.01
Abdominal pain	6.2 ± 2.5	6.1 ± 2.5	−2.5 ± 2.9	−2.9 ± 3.4	0.4	−1.2 to 2.0	0.620	0.13
Abdominal bloating	7.8 ± 2.2	7.4 ± 2.1	−3.2 ± 2.9	−2.5 ± 3.3	−0.7	−2.3 to 0.9	0.373	0.23
Meteorism	6.9 ± 2.8	7.2 ± 2.5	−2.0 ± 3.0	−2.2 ± 3.3	0.2	−1.4 to 1.8	0.789	0.07
Flatulence	6.5 ± 2.9	6.0 ± 2.3	−1.5 ± 3.3	−1.8 ± 5.0	0.3	−1.5 to 2.1	0.759	0.08
Constipation	6.7 ± 3.5	6.1 ± 3.6	−3.2 ± 3.6	−2.6 ± 4.1	−0.6	−2.6 to 1.3	0.522	0.17
Diarrhoea	2.6 ± 3.5	4.6 ± 3.9	−1.2 ± 3.3	−2.5 ± 4.5	1.3	−0.8 to 3.4	0.213	0.34
Nausea	3.3 ± 3.2	3.3 ± 3.3	−2.8 ± 3.1	−2.6 ± 2.7	−0.2	−1.7 to 1.3	0.787	0.07
Vomiting	0.7 ± 1.8	0.8 ± 2.3	−0.2 ± 1.0	−0.5 ± 1.3	0.4	−0.2 to 1.0	0.231	0.31
Belching	4.2 ± 3.2	4.1 ± 3.5	−1.3 ± 2.9	−1.6 ± 2.4	0.3	−1.1 to 1.7	0.678	0.11
Dyspepsia	6.2 ± 3.0	6.5 ± 2.9	−3.2 ± 3.4	−3.8 ± 3.2	0.7	−1.0 to 2.3	0.444	0.20

CI, confidence interval; ETD, estimated treatment difference (positive figures favour VSL#3); FMS, fibromyalgia syndrome; SD, standard deviation.

**Table 4 pharmaceuticals-14-01063-t004:** Comparison of the impact of VSL#3^®^ and placebo on fibromyalgia, sleep, depression and quality of life.

	Baseline (Mean ± SD)		Mean Change (±SD) from Baseline to Week 12		Treatment Difference (Placebo Minus VSL#3)			
Outcome	Placebo N = 56	VSL#3^®^ N = 54	Placebo N = 35	VSL#3^®^ N = 28	ETD	95% CI	*p*-Value	Cohen’s d
FIQR-total	75.5 ± 12.3	70.0 ± 17.8	−12.5 ± 14.1	−7.2 ± 12.5	−5.2	−12.0 to 1.6	0.128	0.40
FIQR-pain	8.0 ± 1.6	7.8 ± 1.6	−0.7 ± 2.0	−0.9 ± 2.3	0.3	−0.8 to 1.3	0.611	0.13
FIQR-energy	7.9 ± 2.4	7.6 ± 2.4	−0.6 ± 3.2	−1.0 ± 2.9	0.4	−1.2 to 1.9	0.635	0.12
FIQR-stiffness	8.1 ± 2.1	7.3 ± 2.5	−1.7 ± 2.9	−0.3 ± 2.3	−1.5	−2.8 to 0.1	0.034	0.56
ISI total	19.9 ± 4.6	17.3 ± 7.0	−1.2 ± 4.0	−1.7 ± 5.9	0.5	−2.0 to 3.0	0.702	0.10
PHQ-9	17.4 ± 5.6	16.3 ± 6.3	−2.5 ± 4.2	−2.2 ± 7.2	−0.3	−3.4 to 2.8	0.846	0.05
SF-36 PCS *	28.4 ± 7.0	27.9 ± 6.3	2.2 ± 6.3	4.5 ± 8.0	−2.3	−5.9 to 1.3	0.211	0.33
SF-36 MCS *	32.1 ± 11.8	32.6 ± 12.8	1.9 ± 12.2	0.8 ± 12.4	1.1	−5.3 to 7.4	0.740	0.09

* The number of observed cases at week 12 was N = 35 and N = 27 for placebo and VSL#3, respectively. CI, confidence interval; ETD, estimated treatment difference (positive figures favour VSL#3); FIQR, Revised Fibromyalgia Impact Questionnaire; ISI, Insomnia Severity Inventory; MCS, Mental Component Score; PHQ-9, 9-item Patient Health Questionnaire; PCS, Physical Component Score; SD, standard deviation; SF-36, Short-Form Health-Survey.

**Table 5 pharmaceuticals-14-01063-t005:** Safety and tolerability profiles of VSL#3^®^ and placebo.

Outcome [N (%)]	Placebo N = 56	VSL#3^®^ N = 54	*p*-Value
At least one adverse event	19 (33.9)	20 (37.0)	0.733
Treatment discontinuation due to adverse events	6 (10.7)	7 (13.0)	0.714
Serious adverse events	0 (0.0)	0 (0.0)	NA
Most frequent adverse events ^a^ (incidence ≥ 3%)			
Abdominal distension	1 (1.8)	5 (9.3)	0.110
Flatulence	3 (5.4)	5 (9.3)	0.490
Abdominal pain	3 (5.4)	3 (5.6)	1.000
Constipation	4 (7.1)	3 (5.6)	1.000
Diarrhoea	0 (0.0)	2 (3.7)	0.240
Vomiting	0 (0.0)	2 (3.7)	0.240
Nausea	3 (5.4)	2 (3.7)	1.000
Disease worsening	0 (0.0)	2 (3.7)	0.240
Dyspepsia	3 (5.4)	0 (0.0)	0.240
Headache	2 (3.6)	0 (0.0)	0.490
Upper abdominal pain	4 (7.1)	0 (0.0)	0.120
Swelling	2 (3.6)	0 (0.0)	0.490
Influenza	2 (3.6)	0 (0.0)	0.490

^a^ Adverse events were coded with the Medical Dictionary for Regulatory Activities (MedRA) and are presented as preferred terms. NA = not applicable as the data did not fulfil the criteria required to perform a Fisher’s test.

## Data Availability

The clinical trial data are available from the corresponding author upon reasonable request.

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
