# Peer review of "The Probiotic VSL#3® Does Not Seem to Be Efficacious for the Treatment of Gastrointestinal Symptomatology of Patients with Fibromyalgia: A Randomized, Double-Blind, Placebo-Controlled Clinical Trial"

_pharmaceuticals, 2021, doi:10.3390/ph14101063_

Round 1

Reviewer 1 Report

The study by Calandre et al. evaluate the efficacy and tolerability of a multi-strain probiotic, VSL#3, for the treatment of fibromyalgia-associated gastrointestinal manifestations. Please see my comments below: 1) Abstract: lack of what specific treatment – please be more explicit here. 2) Abstract: what kinds of trial - Randomised or non-randomised? Also what kinds of active treatment – the authors need to provide sufficient details for the readers to understand the study by just glancing at the abstract. 3) Introduction: Fibromyalgia is a complex syndrome – this sentence does not provide any understanding to the readers to understand this disorder. 4) Line 60-66: this should be in Methods section. This should be replaced with a general introduction to probiotics and why probiotics should be used for this syndrome. Here, the gap needs to be identified. 5) Methods: It is unclear to the readers how the sample size was calculated because the explaination given is vague. Why 110 not other numbers? Also the effect sizes were given. Why is this given here? The questionnaires used were validated, which provided acceptable validity to the study design. 6) The missing data were imputed using LOCF. What is the percentage of missing data in the study? LOCF is not the best imputation method for a RCT. I suggest the authors can drop this analysis and do a sensitivity analysis. Otherwise, the authors should perform a regression imputation for the missing data. 7) Results: How many people were screened for the study? 8) Results should be organised for primary and secondary findings. Moreover, please present the data for participants with no missing data and with missing data, to see if any differences in the main findings in particular. 9) Please also include covariates in the analysis. I would suggest please use ANCOVA instead of t-tests for the analysis. 10) The language and style of the manuscript need to be improved.

Author Response

The study by Calandre et al. evaluate the efficacy and tolerability of a multi-strain probiotic, VSL#3, for the treatment of fibromyalgia-associated gastrointestinal manifestations. Please see my comments below:

We thank the reviewer for her/his comments. Changes in the text are shown in red color

1) Abstract: lack of what specific treatment – please be more explicit here.

2) Abstract: what kinds of trial - Randomised or non-randomised? Also what kinds of active treatment – the authors need to provide sufficient details for the readers to understand the study by just glancing at the abstract.

Following the reviewer’s suggestion, we have performed the requested modifications in the abstract text.

3) Introduction: Fibromyalgia is a complex syndrome – this sentence does not provide any understanding to the readers to understand this disorder.

Following the reviewer’s suggestion, we have modified the text to explain why fibromyalgia is a complex syndrome

4) Line 60-66: this should be in Methods section. This should be replaced with a general introduction to probiotics and why probiotics should be used for this syndrome. Here, the gap needs to be identified.

It seems that the template is not always the same for each person as. In our template, lines 60-66 include the last four lines of one paragraph and the two first lines of the following one. We think that the reviewer refers to the description and composition of VSL#3. If so, this information has been moved to the methods section.

Regarding the rationale for using probiotics in this syndrome, although we thought that we had adequately explained the rationale of the study, we have added more information at this respect.

5) Methods: It is unclear to the readers how the sample size was calculated because the explaination given is vague. Why 110 not other numbers? Also the effect sizes were given. Why is this given here? The questionnaires used were validated, which provided acceptable validity to the study design.

We have added a clarification in relation with the estimated sample size

We are not sure to understand the reviewer`s comment on the effect sizes. We included the calculation of the effect sizes (Cohen’s d) in the Methods section to facilitate the clinical interpretation of the data. As we mention in the Methods section, effect sizes could be interpreted as trivial if <0.2, small if they were between 0.2 and <0.5, medium if they were between 0.5 and <0.8, and large if ≥0.8. Usually, to be interpreted as a clinically relevant, the effect size as measured with the Cohen’s d should be at least medium (i.e. equal or greater than 0.5).

6) The missing data were imputed using LOCF. What is the percentage of missing data in the study? LOCF is not the best imputation method for a RCT. I suggest the authors can drop this analysis and do a sensitivity analysis. Otherwise, the authors should perform a regression imputation for the missing data.

There is no universal method for handling missing data in a clinical trial. The LOCF approach is one of the methods available. A primary analysis can only be accepted if it is considered that an important bias in favor of the experimental treatment can be reasonably excluded. Due to nature of the data, it is unlikely that the LOCF approach have biased the results in favor of the VSL#3. However, following the reviewers suggestion, we have included a sensitivity analysis of the primary outcome and some key secondary outcomes (i.e. gastrointestinal symptoms) using a complete case approach and the results show two findings: first, as expected, the within-group mean changes were larger in this analysis as compared to the LOCF analysis; second, results of the comparisons between placebo and VSL#3 using the complete case approach were generally similar to those observed in the LOCF analyses. The consistency of the results of the LOCF and complete case analyses provides some assurance that neither the missing information nor the methods used to handle missing data had an important effect on the overall study conclusions. Finally, bearing in mind the high placebo response and high drop-out rate in the experimental group, it is very likely that LOCF analysis in our trial was a conservative approach.

7) Results: How many people were screened for the study?

Unfortunately, we did not record the number of patients who were screened and finally were not randomized, although, based on our personal impression, we can state that they were few.

8) Results should be organised for primary and secondary findings Moreover, please present the data for participants with no missing data and with missing data, to see if any differences in the main findings in particular.

As suggested, we have organized the results according to primary and secondary outcomes. Also, as mentioned above, we have included a sensitivity analysis using a complete case approach and the results are generally consistent with those of the primary analysis presented in the manuscript (see Table 2 and the new supplementary table)

9) Please also include covariates in the analysis. I would suggest please use ANCOVA instead of t-tests for the analysis.

We understand the reviewers’ point, but we think that the analysis we have performed is also correct. As we stated in the manuscript, at baseline, study groups were generally well-balanced regarding demographic and clinical characteristics and, more importantly, regarding the individual gastrointestinal symptoms of abdominal pain, abdominal bloating, meteorism and the composite score of those three symptoms. Thus, the mean composite score of pain+bloating+meteorism (i.e. the primary outcome) was 20.9±5.6 in the placebo group and 20.7±5.0 in the VSL#3 group. Therefore, in our view, it is unlikely that an ANCOVA using the baseline score as a covariate would have changed the results and interpretation of this trial

10) The language and style of the manuscript need to be improved

Following the reviewer’s suggestion, the revised version of the manuscript has been edited by a professional English Editing Service.

Reviewer 2 Report

Dear author:

I read up on the type of study and the probiotic product used before reviewing the manuscript in detail.

I went to verify registration NCT04256785. I found it says: "Results information has been submitted to ClinicalTrials.gov by the sponsor or investigator, but is not yet publicly available (or "posted") on ClinicalTrials.gov. The submitted information may not be available if it is pending Quality Control (QC) Review by the National Library of Medicine (NLM) or if issues identified during QC review are being addressed or corrected by the sponsor or investigator".

I do not think it is correct to publish results before there has been a completion of the procedures on clinical trials. Moreover, the authors state in the manuscript, "Actial Farmaceutica Srl reviewed a final draft of this manuscript and provided not binding comments to the authors". This sentence is ambiguous. The authors should submit for transparency the file with the "not binding comments do the authors" provided by Actial (the sponsor).

In addition, from the publicly available information, the VSL#3 product used by the investigators is a different product than that previously sold under the same brand name.

https://schulmanbh.com/news/federal-judge-upholds-18-million-jury-verdict-in-favor-of-schulman-bhattacharyas-clients-enters-broad-permanent-injunction-to-block-further-false-advertising-of-the-vsl3-product/

and

https://www.classaction.org/news/class-action-accuses-pharma-companies-of-secretly-reformulating-vsl3-probiotics-product

and

https://www.mdpi.com/2072-6643/12/2/399

The authors do not inform the reader about these serious legal issues regarding the legacy of VSL#3. The manuscript also uses bibliographic references (e.g., n.17; n.25) that lead to believe a continuity between what was previously published on VSL#3 and the recently marketed VSL#3 product, which seems to be a totally different product. It is necessary that extensive portions of the discussion referring to the formulation previously sold as VSL#3, either in the form of original articles or meta-analysis, be deleted. Basically, according to what was reported (NCT04256785), in 2018, they treated patients with a VSL#3 product without any previous clinical history. This should be made clear.

I would like to point out that I am surprised that the authors have not considered clarifying all that is around VSL#3 product. I cannot accept that incomplete or wrong information is published for two reasons. The first reason is the possible consequences that the journal could face, given that there are several rulings that prohibit creating a continuity between the formulation previously sold under the brand VSL#3 and the new one. Among other things, I am concerned that clinical trials have suspended the publication of the results of the study.

It is my opinion that the authors should clarify the above essential points. Since from what is public, around the VSL#3 product, there are still several legal controversies. I think that the authors should be transparent regarding their possible involvement in such disputes and to what extent.

I will be happy to review the manuscript after the authors have resubmitted it and clarified the points mentioned above.

Author Response

Dear author:

I read up on the type of study and the probiotic product used before reviewing the manuscript in detail.

I went to verify registration NCT04256785. I found it says: "Results information has been submitted to ClinicalTrials.gov by the sponsor or investigator, but is not yet publicly available (or "posted") on ClinicalTrials.gov. The submitted information may not be available if it is pending Quality Control (QC) Review by the National Library of Medicine (NLM) or if issues identified during QC review are being addressed or corrected by the sponsor or investigator". I do not think it is correct to publish results before there has been a completion of the procedures on clinical trials.

Key ethical obligations of the researchers are the registration of research studies and making the results publicly available. We fully adhere to those obligations. Thus, the record NCT04256785 has been publicly available from February 2020. In addition, although it is not mandatory, the study results are also available at Clinicaltrials.gov from July 2021 and they have recently passed the quality control review by NLM. Moreover, we are trying to publish the results as a full paper. Therefore, we think we are fulfilling all ethical obligations

Moreover, the authors state in the manuscript, "Actial Farmaceutica Srl reviewed a final draft of this manuscript and provided not binding comments to the authors". This sentence is ambiguous. The authors should submit for transparency the file with the "not binding comments do the authors" provided by Actial (the sponsor).

Transparency is also an important issue on research publication. In our view, the key issue for the reader is to know what the role in the study of the funders was; in our manuscript that information is fully disclosed including that Actial had the opportunity to review the manuscript and provide comments. In our experience, it is quite unusual to provide the specific comments that any entity could have made to the content of a manuscript. It is also important to bear in mind, that the byline authors are fully responsible for the content of the manuscript. However, if the journal’s, editor consider it relevant to provide the comments made by the Actial, we are willing to do it. 

It is also important to clarify that the sponsor of this study was not Actial. This was an independent study sponsored by the principal investigator, Elena P. Calandre. We have included a clarification on the text of the manuscript. Actial partially funded this research and its role has been already disclosed in the manuscript.

0n addition, from the publicly available information, the VSL#3 product used by the investigators is a different product than that previously sold under the same brand name.

https://schulmanbh.com/news/federal-judge-upholds-18-million-jury-verdict-in-favor-of-schulman-bhattacharyas-clients-enters-broad-permanent-injunction-to-block-further-false-advertising-of-the-vsl3-product/ and https://www.classaction.org/news/class-action-accuses-pharma-companies-of-secretly-reformulating-vsl3-probiotics-product and  https://www.mdpi.com/2072-6643/12/2/399. The authors do not inform the reader about these serious legal issues regarding the legacy of VSL#3.

This is a scientific paper reporting the results of a randomized clinical trial with a probiotic. From our perspective it is important to state the following issues:

-The purpose of the study was to test a probiotic product in fibromyalgia. Among others, we identified VSL#3, a commercially -and legally- available food supplement in Spain, the country where the trial has been performed.

-The composition of VSL#3 product used in our trial is already described in the manuscript.

-The trial was conducted fulfilling all the Spanish regulations on clinical trials, and, as it is already stated in the manuscript, following the international ethical guidelines on clinical research.

-The role of the funders and the potential conflict of interest are appropriately disclosed in the manuscript.

Therefore, we think we are providing all the scientific information a reader needs to know to critically read and judge our study.

A different issue is the legal dispute between the distributors of the product VSL#3® and Professor Claudio De Simone, that the reviewer mentions and cites in his/her comment. In our view, a scientific paper is not the right context where discuss about such a complex legal matter. Our role as researchers is to answer clinically relevant questions.

The manuscript also uses bibliographic references (e.g., n.17; n.25) that lead to believe a continuity between what was previously published on VSL#3 and the recently marketed VSL#3 product, which seems to be a totally different product. It is necessary that extensive portions of the discussion referring to the formulation previously sold as VSL#3, either in the form of original articles or meta-analysis, be deleted.

The references we cite in the manuscript have been published in peer reviewed journals and explicitly mention VSL#3 as the product they were evaluating.  However, taking into account the comment from the reviewer, we have included a clarification (written in green colour) on the discussion regarding the potential differences between the VSL#3 formulation used in our study and the formulation used in studies conducted prior to 2016 and therefore mentioned in those references. We have also added the following reference to support that comment: de Simone C. Comment on: "Search and Selection of Probiotics that Improve Mucositis Symptoms in Oncologic Patients: A Systematic Review. Nutrients 2019, 11, 2322". Nu-trients. 2020 Feb 3;12(2):399. doi: 10.3390/nu12020399. PMID: 32028554; PMCID: PMC7071258.

Basically, according to what was reported (NCT04256785), in 2018, they treated patients with a VSL#3 product without any previous clinical history. This should be made clear.

Regarding the reviewer’s statement “they treated patients with a VSL#3 product without any previous clinical history”, we are not sure to understand what the reviewer’s point is. If the reviewer is trying to indicate that there was not rationale for testing VSL#3 in this clinical condition, we think this manuscript provides that rationale. And the rationale included in the protocol was considered adequate by the Ethics Committee that reviewed and approved the trial. The formulation of VSL#3 we have used is commercially available in Spain since June 2017.

I would like to point out that I am surprised that the authors have not considered clarifying all that is around VSL#3 product. I cannot accept that incomplete or wrong information is published for two reasons. The first reason is the possible consequences that the journal could face, given that there are several rulings that prohibit creating a continuity between the formulation previously sold under the brand VSL#3 and the new one. Among other things, I am concerned that clinical trials have suspended the publication of the results of the study.

As states several times in response to this reviewer, we are providing all the scientific information surrounding this trial. As we stated above, we have included a comment on the discussion regarding the potential differences between the VSL#3 formulation used in our study and the formulation used in studies conducted prior to 2016

In addition, we have to clearly state that the reviewer’s concern regarding the publication of the results by clinicaltrials.gov, in addition to be unpolite and improper from the scientific point of view, is absolutely unjustified. See our previous response to the issue of clinicaltrial.gov.

It is my opinion that the authors should clarify the above essential points. Since from what is public, around the VSL#3 product, there are still several legal controversies. I think that the authors should be transparent regarding their possible involvement in such disputes and to what extent.

We have nothing to add to what has been extensively explained in previous responses.

However, we would like to cite here the first sentences of the preface of an excellent book “On Being a Scientist: Responsible Conduct in Research”: “The scientific research enterprise, like other human activities, is built on a foundation of trust. Scientists trust that the results reported by others are valid. Society trusts that the results of research reflect an honest attempt by scientists to describe the world accurately and without bias” (https://www.nap.edu/catalog/4917/on-being-a-scientist-responsible-conduct-in-research-second-edition). We are scientists and as such we have tried to honestly report the design and results of our research. The scientific merit should be judged by the reviewers and eventually the readers.

I will be happy to review the manuscript after the authors have resubmitted it and clarified the points mentioned above.

Reviewer 3 Report

The authors aim at verifying the ability of VSL#3â probiotics to alleviate gastrointestinal symptoms in patients with fibromyalgia. The study lacks of novelty and it is not properly conceived.

A major problem is the absence of a control untreated group.

The two groups are balanced regarding demographic and clinical characteristics. However, considering that several subjects didn’t complete the study, resulting in 35 subjects in the placebo group and 28 subjects in the VSL#3® group, the group size should be increased.

Data on microbiota composition before the treatment, after the treatment and after the follow-up period could be useful to discuss clinical observations.

Author Response

We thank the reviewer for her/his comments. Changes in the text are shown in blue colour.

The authors aim at verifying the ability of VSL#3â probiotics to alleviate gastrointestinal symptoms in patients with fibromyalgia. The study lacks of novelty and it is not properly conceived.

We are sorry to know that the reviewer find that our study lacks novelty. This is the second study that investigates the effects of a probiotic in the treatment of fibromyalgia (the former one evaluated the effects on mood and cognition) and, to our knowledge, the first one to evaluate any kind of treatment for the gastrointestinal symptoms associated to fibromyalgia. Therefore, we think that our study, albeit exploratory, adds new information on the management of this complex disease and the potential role of probiotics

A major problem is the absence of a control untreated group.

We think that the key issue is what is the best control group for evaluating the effect of a probiotic on these symptoms. We think that a placebo is the most appropriate. The use of placebo allows us to control for the non-specific effect of the interventions and also to control for the natural course of the disease.  A control untreated group, theoretically speaking, would be adequate for controlling the natural course of the disease, but not for controlling the non-specific effect of the interventions (e.g. patients’ expectations). Bearing in mind the high placebo response in our trial, we think it is very likely that a trial comparing the probiotic with an untreated group would have overestimated the benefit of the probiotic. Adding a third control group on untreated patients would have been unfeasible for us regarding sample size and would have prevented the use of an appropriate masking.

The two groups are balanced regarding demographic and clinical characteristics. However, considering that several subjects didn’t complete the study, resulting in 35 subjects in the placebo group and 28 subjects in the VSL#3® group, the group size should be increased.

We recognize that we had an high number of dropouts and this is mentioned as one of the study limitations. However, it is not possible to increase the number of patients included in a clinical trial “post hoc”. Please, also note that this was an exploratory trial

Data on microbiota composition before the treatment, after the treatment and after the follow-up period could be useful to discuss clinical observations.

We acknowledge that the analysis of microbiota would have been an added value to the study. We have added the lack of this analysis to the study limitations.

Round 2

Reviewer 1 Report

Please see my comments below:

1) I would like to suggest the authors to remove the LOCF analysis and only report complete case. The study had a high dropout, which did not favour the use of data imputation.

2) The authors did not explain how they calculated the 110 patients in their sample size.

3) Please add this recent reference to these sentences "Probiotics have been studied in a variety of 63
clinical conditions, including gastrointestinal disorders, dermatological disorders and 64
metabolic diseases [15]. Among gastrointestinal disorders, the most studied condition is 65
IBS, where probiotics seem to exert a favourable response in global symptoms, although 66
there are not still enough data to specify which individual probiotic could be more effec- 67
tive. Probiotics seem to be, in general, well tolerated " :

Yusof, N., Hamid, N., Ma, Z.F. et al. Exposure to environmental microbiota explains persistent abdominal pain and irritable bowel syndrome after a major flood. Gut Pathog 9, 75 (2017). https://doi.org/10.1186/s13099-017-0224-7

Author Response

Reviewer 1:

We thank the reviewer for her/his comments; as before, changes in the text are written in red colour

1) I would like to suggest the authors to remove the LOCF analysis and only report complete case. The study had a high dropout, which did not favour the use of data imputation.

We do agree that the high drop-out rate is an important issue and it is recognized as such among the study limitations. However, in our view, presenting the complete case analysis as the single -and primary- analysis is an approach that overestimates treatment effects as compared to the LOCF approach. According to Hughes et al. (2019)*, complete case analysis is appropriate only when the participants in the analysis can be regarded as a random sample of the study population (i.e., when the missing mechanism is missing completely at random “MCAR”), which is not the case in our study. Therefore, despite the negative impact of the dropouts in the current LOCF approach, we think it is more conservative than the complete case analysis.

In summary, what we suggest and have done is to maintain the LOCF analysis and to move the table of the complete case analyses from the supplementary information to the main body of the manuscript (presenting these results as a sensitivity analysis). In addition, and consistently with the reviewer’s comment, we have stressed on the potential impact of the dropouts in the efficacy analysis. Thus, the following sentence of the study limitations has been amended:

It read: “The dropout rates as well as the placebo effect in our study were relatively high, prompting careful interpretation of the study findings”

Currently, it reads: The high dropout rate and its impact on the study estimates because of the missing data, as well as the high placebo effect in our study, prompt cautious interpretation of the study findings

* Hughes RA, Heron J, Sterne JAC, Tilling K. Accounting for missing data in statistical analyses: multiple imputation is not always the answer. Int J Epidemiol. 2019 Aug 1;48(4):1294-1304. doi: 10.1093/ije/dyz032. PMID: 30879056; PMCID: PMC6693809.

2) The authors did not explain how they calculated the 110 patients in their sample size.

As mentioned at the beginning of the statistical analysis section, it was not possible to carry out the sample size calculation due to the absence of previous evidence, via interventional studies, on the efficacy of probiotics on the gastrointestinal symptoms (i.e., our primary outcome) in fibromyalgia. Therefore, the sample size was solely based on the feasibility of recruitment taking into consideration the volume of patients attending each of the two participating centers. This has been further clarified in the text under section “4.5. Statistical analysis”.

3) Please add this recent reference to these sentences "Probiotics have been studied in a variety of 63clinical conditions, including gastrointestinal disorders, dermatological disorders and 64metabolic diseases [15]. Among gastrointestinal disorders, the most studied condition is 65 IBS, where probiotics seem to exert a favourable response in global symptoms, although 66 there are not still enough data to specify which individual probiotic could be more effec- 67 tive. Probiotics seem to be, in general, well tolerated " :

Yusof, N., Hamid, N., Ma, Z.F. et al. Exposure to environmental microbiota explains persistent abdominal pain and irritable bowel syndrome after a major flood. Gut Pathog 9, 75 (2017). https://doi.org/10.1186/s13099-017-0224-7

We have now cited the mentioned reference in the text.

Reviewer 2 Report

The study by Calandre and colleagues evaluate the efficacy and tolerability of a multi-strain probiotic, VSL#3, for the treatment of fibromyalgia-associated gastrointestinal manifestations. A careful evaluation of the submitted manuscript revealed some major and minor issues that have to be corrected in order to make the study publishable:

Major revisions

Line 31: since the submitted study substantially presents the lack of differences between the treated group and the placebo one, the sentence “Although our results suggest that some patients could benefit from treatment with VSL#3” results to be inappropriately inserted in the conclusion of the abstract. In order to better understand the importance of such a thing, one could affirm that “Although our results suggest that some patients could be damaged by the treatment with VSL#3” since some parameters seems to be worsened by the administration of the VSL#3 product. Please remove such a sentence or rephrase the paragraph.

Line 85: in figure 1 it is reported that 7 subjects in the placebo group and 7 subjects in the VSL#3® one were lost to follow-out while they further declared that “All randomised subjects were included in the safety analysis”. Authors should explain such apparent discrepancy or modify the sentence.

Lines 208-210: since the consent withdraw represents a critical concern, especially with respect to the tolerability or safety of treatments, authors should be more precise in providing information about the reason leading some patients to abandon the study. I encourage authors to provide a supplementary file with detailed descriptions about the exclusion of patients.

Minor revisions

Line 65-67: References are needed for supporting the sentence “Among gastrointestinal disorders, the most studied condition is IBS, where probiotics seem to exert a favourable response in global symptoms, although there are not still enough data to specify which individual probiotic could be more effective.”

Line 84: Please replace PGI with Patient Global Improvement Scale (PGI) since it is the first time it has been introduced in the manuscript.

Line 97: the sentence “at baseline, study groups were generally well-balanced” lacks specification. Even if I understand the colloquial intent of authors, I strongly recommend clarifying that the studied groups were “homogeneous” or “non statistically different” adding calculated p values in table 1.

Lines 171-177 Authors should clarify if statistically significant differences were present or not between groups since such information seem not to be reported. In order to make the results more comprehensible to the readers, I strongly recommend adding calculated p values to table 4.

Line 222: Authors should explain the concept behind the “Global IBS symptom score” since it was never mentioned before in the text and the use of validated assays for IBS has not been reported in the manuscript.

Line 231: please remove the word “real” in the sentence, since the use of such word postulate the not permitted existence of not real benefits.  

Lines 239-240: Authors should indicate which references are in agreement with the sentence “With one exception, they also agree in considering that multi-strain probiotics seem to be preferable over single-strain probiotics”.

Line 256: please replace “VSL#” with VSL#3

Lines 259-262: Authors have to specify reference(s) relative to the sentence “We would like to note that some authors have reported that the formulation of VSL#3 used in the studies conducted prior to 2016 is not the same as the one used and thus, the results reported in this meta-analysis could be referred to that formulation and not the one we used in our study.”

Line 263: Authors should briefly list the clinical conditions considered in the cited meta-analysis.

Line 291: Since the VSL#3 product is currently not sold in Spain by Ferring, authors should specify who currently sell the product on the Spanish market or that the product was sold in the past but not currently.

Line 323: please add “then” between other and IBS. Authors should clarify which information have been used to determine the enrolled patients suffering from IBS.

Line 355: please replace Patient Global Improvement Scale (PGI)

Line 361: Authors should provide information about the composition of the given placebo

Author Response

Reviewer 2:

We thank the reviewer for her/his comments; changes in the text are written in green colour

The study by Calandre and colleagues evaluate the efficacy and tolerability of a multi-strain probiotic, VSL#3, for the treatment of fibromyalgia-associated gastrointestinal manifestations. A careful evaluation of the submitted manuscript revealed some major and minor issues that have to be corrected in order to make the study publishable:

Major revisions

Line 31: since the submitted study substantially presents the lack of differences between the treated group and the placebo one, the sentence Although our results suggest that some patients could benefit from treatment with VSL#3results to be inappropriately inserted in the conclusion of the abstract. In order to better understand the importance of such a thing, one could affirm that Although our results suggest that some patients could be damaged by the treatment with VSL#3since some parameters seems to be worsened by the administration of the VSL#3 product. Please remove such a sentence or rephrase the paragraph.

We have revised the conclusions in the abstract deleting the sentence mentioned by the reviewer and indicating instead the lack of differences between the two groups at the level of the primary and secondary outcomes.

Line 85: in figure 1 it is reported that 7 subjects in the placebo group and 7 subjects in the VSL#3® one were lost to follow-out while they further declared that All randomised subjects were included in the safety analysis. Authors should explain such apparent discrepancy or modify the sentence.

The safety population comprised all patients who received at least one dose of the study products. This is a standard definition of safety population in clinical trials. In our study all randomized patients received at least one dose of the study product and therefore were included in the safety population. The assessment of potential adverse events was done in these subjects who were lost to follow up on each of the scheduled visits, as per the study protocol, until the loss of follow-up. This procedure is followed in most randomized clinical trials We have provided further clarification on the safety analysis set in figure 1.

Lines 208-210: since the consent withdraw represents a critical concern, especially with respect to the tolerability or safety of treatments, authors should be more precise in providing information about the reason leading some patients to abandon the study. I encourage authors to provide a supplementary file with detailed descriptions about the exclusion of patients.

The whole picture of the patient disposition is already presented in figure 1. Regarding patients who withdrew consent (1 among placebo-treated patients and 4 among VSL#3-treated patients) we have no further information. It is important to bear in mind that patients -as it is stated in the informed consent- have the right to discontinue their participation in the study without providing any kind of explanation to the investigator. Patient’s decision could be related with tolerability/safety issues, but also with efficacy issues or being unrelated with the study product.

Minor revisions

Line 65-67: References are needed for supporting the sentence Among gastrointestinal disorders, the most studied condition is IBS, where probiotics seem to exert a favourable response in global symptoms, although there are not still enough data to specify which individual probiotic could be more effective.

This statement is supported by reference 17 that was inserted in the text at the end of the paragraph; we have now also inserted it at the end of the sentence.

Line 84: Please replace PGI with Patient Global Improvement Scale (PGI) since it is the first time it has been introduced in the manuscript.

The requested modification has been done.

Line 97: the sentence at baseline, study groups were generally well-balancedlacks specification. Even if I understand the colloquial intent of authors, I strongly recommend clarifying that the studied groups were homogeneous” or “non statistically differentadding calculated p values in table 1.

This is a randomized study and, as such, comparison of the baseline characteristics does not provide relevant information. If properly randomized, any difference in baseline characteristic is due to chance (see references below). Moreover, baseline comparison might be somewhat misleading; a difference that is not statistically significant could be clinically relevant if the variable is a strong prognostic factor. Therefore, the judgment of comparability of baseline characteristic is subjective and based on clinical grounds.

Altman AR. Comparability of randomised groups. Statistician. 1985;34:125–36.

Pocock SJ, Assmann SE, Enos LE, Kasten LE. Subgroup analysis, covariate adjustment and baseline comparisons in clinical trial reporting: current practice and problems. Stat Med. 2002;21:2917–30.

Lines 171-177 Authors should clarify if statistically significant differences were present or not between groups since such information seem not to be reported. In order to make the results more comprehensible to the readers, I strongly recommend adding calculated p values to table 4.

The p-values have been calculated and added to the table.

Line 222: Authors should explain the concept behind the Global IBS symptom scoresince it was never mentioned before in the text and the use of validated assays for IBS has not been reported in the manuscript.

The requested clarification has been added to the text.

Line 231: please remove the word realin the sentence, since the use of such word postulate the not permitted existence of not real benefits.  

The word “real” has been removed.

Lines 239-240: Authors should indicate which references are in agreement with the sentence With one exception, they also agree in considering that multi-strain probiotics seem to be preferable over single-strain probiotics”.

The exception has been referenced.

Line 256: please replace VSL#with VSL#3

The replacement has been done.

Lines 259-262: Authors have to specify reference(s) relative to the sentence We would like to note that some authors have reported that the formulation of VSL#3 used in the studies conducted prior to 2016 is not the same as the one used and thus, the results reported in this meta-analysis could be referred to that formulation and not the one we used in our study.

A reference related with this statement has been included.

Line 263: Authors should briefly list the clinical conditions considered in the cited meta-analysis.

The clinical conditions have been detailed

Line 291: Since the VSL#3 product is currently not sold in Spain by Ferring, authors should specify who currently sell the product on the Spanish market or that the product was sold in the past but not currently.

By the time this study was initiated Ferring marketed VSL#3; this laboratory discontinued the marketing of this product by July 2021. However, we think that the relevant information for this report is who manufactures the product and who provided the product for the study, a fact that is indicated in the funding section. On the other hand, for this report, we have already included what we consider relevant information regarding the partial funding Ferring provided us in support of the study activities (data management and statistical analysis). All that information is already provided. We have deleted the sentence “marketed in Spain by Ferring SAU” from the text.

Line 323: please add thenbetween other and IBS. Authors should clarify which information have been used to determine the enrolled patients suffering from IBS.

The word “than” has been inserted and the requirement for all of the mentioned diagnoses has been detailed

Line 355: please replace Patient Global Improvement Scale (PGI)

We think that it is important to expand the names of each scale under the Study assessments section. Therefore, we have deleted the abbreviation and kept the expanded description of the Patient Global Improvement Scale.

Line 361: Authors should provide information about the composition of the given placebo

The composition of the placebo has been detailed in the text

Reviewer 3 Report

The authors answered all the quetions. Considering that limitations are declared and that the microbiota will not be analysed, they should almost cite some papers describing the effects of musti strain probiotic formulations on gut microbiota composition in animal and human studies.

Author Response

Reviewer 3:

We thank the reviewer for her/his comments; as before, changes in the text are written in blue colour

The authors answered all the questions. Considering that limitations are declared and that the microbiota will not be analysed, they should almost cite some papers describing the effects of musti strain probiotic formulations on gut microbiota composition in animal and human studies.

As requested, we have added four references relative to the beneficial changes induced by multi-strain probiotics in gut microbiota and animal and human health.

Round 3

Reviewer 1 Report

After a thorough examination of the manuscript and results, I think the authors still do not asnwer the queries satisfactorily as there are some major flaws in the statistical analysis as mentioned previously.

Author Response

We are sorry to know that the reviewer thinks that we have not answered her/his queries, as we have tried to do it as best as possible. At this respect, we have taken in account the comments of the academic editor and, according to them, we have performed the following changes:

Regarding the issues with the statistical analysis, we have added the following information and Hughes et al reference to the discussion of the study limitations: “Although we performed a secondary analysis using a complete case approach to limit the influence of the imputation method to handle missing data, it is important to bear in mind that complete case analysis is appropriate only when the participants in the analysis can be regarded as a random sample of the study population (i.e., when the missing mechanism is missing completely at random) [32], which cannot be assumed to be the case in our study; in addition, complete case analysis tends to overestimate treatment effects. Therefore, complete case analysis can only be considered a sensitivity analysis”.

The title has been amended for reflecting the negative results of the study. Currently reads: “The Probiotic VSL#3® Does not Seem to be Efficacious for the Treatment of Gastrointestinal Symptomatology of Patients with Fibromyalgia: A Randomized, Double-blind, Placebo-controlled Clinical Trial”

Reviewer 2 Report

In my opinion, the MS could be accepted after a further minor revision (see below).

I have read Panetta's article (ref 19 of the article). Panetta's meta-analysis is based on 120 patients (31 treated with placebo and 89 with VSL#3). 120 is a tiny number for a meta-analysis, considering the extreme heterogeneity of the pathologies (IBS, ulcerative colitis, obesity) even merged with early menopause, all treated with different dosages of VSL#3 for different periods. At the same time, I found on clinical trials, regarding the PROBONE study, in the section results, written by the investigator: 

"35 individuals consented to participate. The investigational product became the subject of a court judgment upholding false advertising claims concerning the product's similarity to earlier products sold under the same name. Because of questions raised by this ruling, no participants are considered as being assigned to a protocol-defined study arm" 

Therefore, if the 17 placebo-treated and 18 VSL#3-treated subjects of the PROBONE study are subtracted, the updated numbers are 14 (placebo arm), and 71 (VSL#3 arm) for a total of 85 individuals left. There can be no doubt that the meta-analysis is indefensible, and the Authors made a serious mistake. Surprisingly, the editor of the journal publishing the article did not check. However, because it is the moral duty of the scientist to be rigorous and ascertain the facts, this journal must not publish the article if the sentence is not deleted or modified. I propose that the Authors remove the phrase and the reference (it does not add anything to the discussion of the results). Alternatively, the authors should write: "There are uncertainties about this meta-analysis because the actual number of patients examined is too small and the pathologies and dosages too heterogeneous."

Author Response

As suggested, the following sentence has been added: “However, there are uncertainties about this meta-analysis because the actual number of patients examined is too small and the pathologies and the probiotic dosages are too heterogeneous”.
